# Pathobiology of *Candida auris* infection analyzed by multiplexed imaging and single cell analysis

Chrystal Chadwick[1], Magdia De Jesus[2,3]*, Fiona Ginty[1], Jessica S. Martinez[1]*

**1** GE Research, Niskayuna, New York, United States of America, **2** Department of Biomedical Sciences, School of Public Health, University at Albany, Albany, New York, United States of America, **3** Division of Infectious Diseases, Wadsworth Center, New York State Department of Health, Albany, New York, United States of America

* jessica.s.martinez@ge.com (JSM); magdia.dejesus@pfizer.com (MDJ)

**Data Availability Statement:** We have made our data available through Zenodo, a stable repository for data, and protocols through protocol io as follows: Cell DIVE™ Platform Slide Clearing and Antigen Retrieval (dx.doi.org/10.17504/protocols.

## Abstract

Fungal organisms contribute to significant human morbidity and mortality and *Candida auris* (*C. auris*) infections are of utmost concern due to multi-drug resistant strains and persistence in critical care and hospital settings. Pathogenesis and pathology of *C. auris* is still poorly understood and in this study, we demonstrate how the use of multiplex immunofluorescent imaging (MxIF) and single-cell analysis can contribute to a deeper understanding of fungal infections within organs. We used two different neutrophil depletion murine models (treated with either 1A8—an anti-Ly6G antibody, or RB6-8C5—an anti-Ly6G/Ly6C antibody; both 1A8 and RB6-8C5 antibodies have been shown to deplete neutrophils) and compared to wildtype, non-neutropenic mice. Following pathologist assessment, fixed samples underwent MxIF imaging using a *C. albicans* antibody (shown to be cross-reactive to *C. auris*) and immune cell biomarkers—CD3 (T cells), CD68 (macrophages), B220 (B cells), CD45 (monocytes), and Ly6G (neutrophils) to quantify organ specific immune niches. MxIF analysis highlighted the heterogenous distribution of *C. auris* infection within heart, kidney, and brain 7 days post-infection. Size and number of fungal abscesses was greatest in the heart and lowest in brain. Infected mice had an increased count of CD3+, CD68+, B220+, and CD45+ immune cells, concentrated around *C. auris* abscesses. CD68+ cells were predominant in wildtype (non-neutropenic mice) and CD3+/CD45+ cells were predominant in neutropenic mice, with B cells being the least abundant. These findings suggest a Th2 driven immune response in neutropenic *C. auris* infection mice models. This study demonstrates the value of MxIF to broaden understanding of *C. auris* pathobiology, and mechanistic understanding of fungal infections.

## Introduction

*Candida auris* (*C. auris*) was first detected in 2009, from an ear canal of a patient in Japan [1]. Since then, the number of *C. auris* infection cases have increased dramatically and have been

io.bpwumpew), Cell DIVE™ Platform Antibody Characterization for Multiplexing (dx.doi.org/10.17504/protocols.io.bpyxmpxn), and Cell DIVE™ Platform Antibody Staining & Imaging(dx.doi.org/10.17504/protocols.io.bpv6mn9e.

**Funding:** The author(s) received no specific funding for this work.

**Competing interests:** The authors have declared that no competing interests exist.

reported around the world (over 20 countries), including the United States of America [2]. *C. auris* has become a global public health challenge and 3–5% of isolates identified world-wide, exhibit pan-resistance that includes resistance to the azoles, polyenes and most importantly the echinocandins, a third class of antifungals currently used to clinically treat resistant *C. auris* [3]. Unlike other Candida species, which typically colonize the intestinal tract, *C. auris* colonizes the skin and can easily transmit it via surface contact, rendering it as a nosocomial pathogen in hospital and nursing home settings [4–7]. Once *C. auris* has entered the bloodstream, fatality rates can be as high as 30–60%. With increasing length of hospital stays in the intensive care unit due to the ongoing COVID-19 pandemic, the concern for this pathogen's transmissibility has greatly increased [8–10]. Most pressing, the pathobiology of *C. auris* remains to be further elucidated and possess challenges in development of alternative treatments and prevention strategies/technologies.

To expand our mechanistic understanding of *C. auris* infection pathology, this study used multiplex immunofluorescent imaging (MxIF) and single cell analysis to characterize the heterogenous immune response of *C. auris*. MxIF iteratively images 60+ protein biomarkers in a single fixed tissue section with conserved spatial and cellular resolution to provide insight into heterogenous cellular environments [11, 12] MxIF has been used extensively in the analysis of tumor and tumor microenvironment [13–18], most recently it has been used to characterize viral (monkeypox) infection response in the epithelium and immune response [19]. Single cell analysis of proteins and associated cell types provides deeper insights into micro and macro-response to infection. MxIF also has potential use in multimodal imaging (i.e. the combination of different imaging techniques and imaging scales) and the development and validation of non-invasive imaging approaches to investigate fungal infection pathology and host response [20]. In this study, we demonstrate the use of MxIF to analyze *C. auris* infected tissues from kidney, heart, and brain in a neutropenic mice model. The neutropenic mice models were developed by treatment with anti-Ly6G antibody (1A8) or anti-Ly6G/Ly6C antibody (RB6-8C5), both shown to deplete neutrophils [21–25]. A wildtype, non-neutropenic model was also used to assess the effect of *C. auris* infection.

## Materials and methods

### Neutrophil depletion murine models and *C. auris* infection

In this study, histological slides generated in Torres et al. [25], where animal studies were conducted under IACUC approved procedures. A brief summary of previous study procedures as follows: neutropenic mice were developed using 8- to 12-week old BALB/c mice (Taconic Farms, Hudson, NY), which were given 200µg through intraperitoneal injections of 1A8 (Bio X Cell, West Lebanon, NH), an anti-Ly6G antibody, or RB6-8C5 (Bio X Cell, West Lebanon, NH), an anti-Ly6G/Ly6C antibody 24 hours prior to *C. auris* infection. Mice were grouped into two phenotypes: non-neutropenic (no treatment or placebo: phosphate buffer saline (PBS)), and neutropenic (treatment with either 1A8 or RB6) (see Table 1).

**Table 1. Outline of phenotypic conditions studied.**

| Phenotype (mice) | Antibody Treatment | Fungal Infection |
|---|---|---|
| Wildtype, non-neutropenic | None (placebo; PBS) | None |
| Wildtype, non-neutropenic | None (placebo; PBS) | *C. auris* |
| Neutropenic | 1A8 | *C. auris* |
| Neutropenic | RB6 | *C. auris* |

Both phenotypes were infected with *C. auris* strain CAU-09 (S. Asian Clade I) and clinical strain M5658 (S. Asian Clade I), acquired from the Centers for Disease Control and Prevention (CDC). Inoculum of 1x10$^7$ cells was administered intravenously (i.v.). The S. Asian clade was used for infections as it was previously found to be less aggregative than other *C. auris* strains [25], which provided ease of counting by a hemacytometer and preparations of dilutions for tail vein infections. Clade I was the major genotype (>99%) among clinical isolates reported from New York, where the initial outbreaks were identified in the United States [26]. Histological slides utilized in this study were from mice sacrificed on day 7 after *C. auris* infection.

## Tissue histology evaluation and sample selection for MxIF

Multi-organ histological slides containing serially sliced sections of brain, lung, stomach, liver, kidney, spleen, and heart from mice with treatment/phenotype conditions outlined in Table 1 were obtained from previous experiments [25]. These multi-organ histological slides underwent multiplex immunohistochemistry (MxIF) processing [15, 27], see Fig 1A. Organs found to have high *C. auris* fungal burden (large presence of fungal abscesses) were selected for further MxIF imaging and analysis, see Fig 1B. Organ tissue slides selected for MxIF analysis were scored for positive *C. auris* cells and evaluated for the presence of immune cells: monocytes (CD68), lymphocytes (CD45), T cells (CD3), B cells (CD45R/B220), and neutrophils (Ly6G) as previously described [25].

## Multiplex Immunohistochemistry (MxIF) of control and infected tissues

Heart, kidney, and brain from the neutropenic infected mice and non-neutropenic/infected and placebo/non-infected controls (Table 1) originally processed for immunohistochemistry

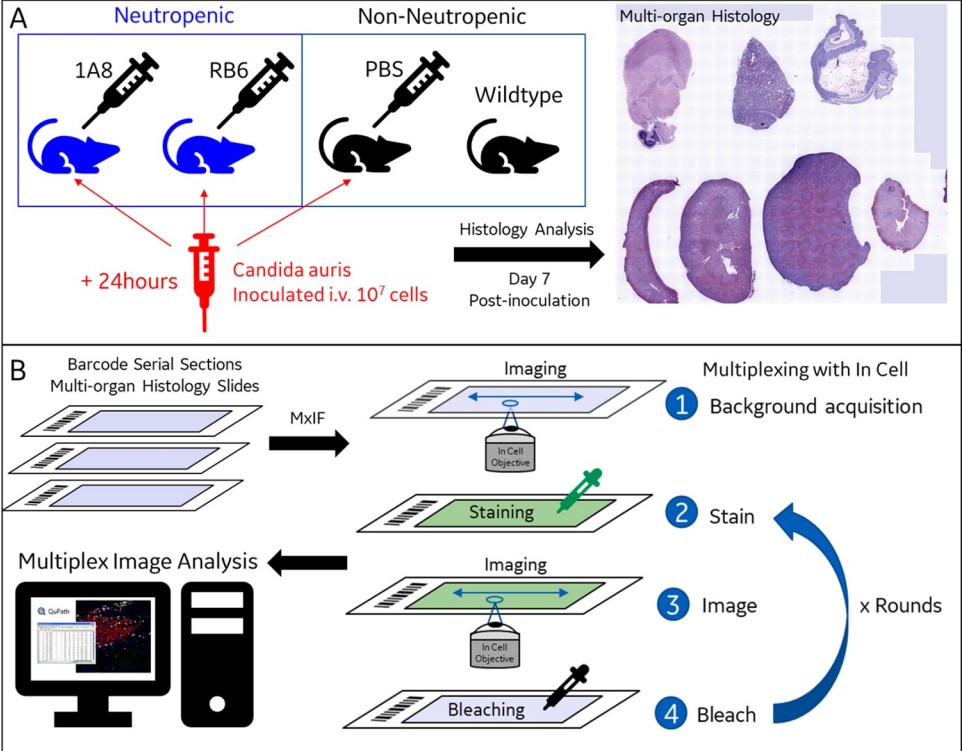

**Fig 1. *Multiplex Immunohistochemistry (MxIF) of neutropenic C. auris infection.*** Neutropenic and non-neutropenic (control, wildtype) mice were infected with *C. auris* (10$^7$ cells) and multi-organ histological slides, serially sectioned, were obtained from these animals (A) and processed for MxIF imaging and analysis (B).

(IHC) in Torres et. al. [25] were further processed for MxIF analysis. Slides were cleared and processed for antigen retrieval as previously described [15]. Protocols (see S1–S3 Files) are also described in protocols.io: Cell DIVE™ Platform Slide Clearing and Antigen Retrieval (dx.doi.org/10.17504/protocols.io.bpwumpew), Cell DIVE™ Platform Antibody Characterization for Multiplexing (dx.doi.org/10.17504/protocols.io.bpyxmpxn), and Cell DIVE™ Platform Antibody Staining & Imaging(dx.doi.org/10.17504/protocols.io.bpv6mn9e).

Briefly, paraffin tissue sections were baked overnight and then taken through a series of xylene and ethanol steps to remove the paraffin. This was followed by a two-step antigen retrieval and slides were blocked overnight in 4% BSA in PBS at 4˚C. After blocking, slides were stained with DAPI (1 μg/ml), cover-slipped with anti-fade mounting media, and background images collected in DAPI, FITC, Cy3 and Cy5 filter channels to record native autofluorescence ("background acquisition").

Afterwards, slides were iteratively stained with antibodies conjugated with fluorescent dyes, imaged, and undergo dye inactivation ("fluorescent dye bleaching") to generate multiplex images. Imaging was conducted on a GE Healthcare IN Cell 2000 with Cell DIVE software (Leica Microsystems) for image registration, autofluorescence subtraction and correction, see Fig 1B. For the initial imaging step, a 10X objective was used, and the whole tissue imaged for DAPI, followed by image stitching to create a composite image of the sample. The image was then converted to a virtual-H&E image (autofluorescence and DAPI converted to color scheme of traditional H&E), which was used for identifying specific regions of interest (ROI) for further analysis (Fig 1A). A minimum of 9 regions with >10 fields were chosen for each histological section, with fungal abscesses serving as the target histological criterial for ROI selection.

Slides were prepared for staining and washing using a Leica Bond autostainer. Antibodies used in this study, underwent a multistep validation process to evaluate specificity and sensitivity of the primary, secondary antibodies and direct conjugates (which are required due to species cross-reactivity) on a multi-organ mouse tissue array, (Pantomics, AMS543 C3H mouse TMA), which included kidney, heart, and brain.

The following antibodies were assessed: Ly6G (Novus Cat No. NBP1-06691), CD45 (Cell Signaling Cat No. 70857S), CD3 (Abcam Cat No. ab16669), CD68 (Abcam Cat No. ab31630), and CD45R/B220 (BD Biosciences Cat No. 557685). Antibody stock and working concentration, dye (fluor), and imaging exposure time optimized for each antibody is outlined in Table 2.

Staining patterns were compared to vendor data sheets and published data and reference spleen and thymus tissues were used to confirm positive signal. We also confirmed that target antigens were not altered by dye inactivation steps by comparing staining on samples that were untreated or treated ~5 times with the dye inactivation process used in MxIF as previously described [15]. Antibodies used in this study were found to be suitable for MxIF analysis at manufacturer's recommended concentrations, see Table 2.

*C. auris* was also stained using a commercially available anti-Candida albicans polyclonal antibody (Thermo Fisher Cat No. PA1-7206), which Torres et al found to cross-react with *C. auris* in culture and in tissues [25]. This was stained with previously described primary/secondary antibody staining protocols [15] and staining performance for conjugates was compared to the primary/secondary antibody used in our antibody validation process.

## Image and statistical analysis of multiplexed images

ImageJ [28] and QuPath [29], both open source software for image processing and bioimage analysis, were used to analyze *C. auris* abscesses within tissues and quantify immune cell types. In the infection groups, regions of interest (ROIs) were selected based on presence of fungal

**Table 2. List of antibodies, dye, and working concentration used in MxIF analysis.**

| Antibody Target | Antibody Vendor; Catalog Number; Clone | Dye | Secondary Antibody Used | Stock Concentration (µg/mL) | Working Concentration (µg/mL) |
|---|---|---|---|---|---|
| *C. auris* | *C. albicans* Thermo Fisher: Cat No. PA1-7206; Polyclonal; demonstrated to be cross reactive with *C. auris* [25] | Cy3 | Donkey Anti-Rabbit Cy3 (5ug/ml) | 4000 | 1 |
| Ly6G | Novus; Cat No.NBP1-06691; EPR22909-135 | Cy5 | Donkey Anti-Rat A647 (5ug/ml) | 1000 | 10 |
| CD45 | Cell Signaling: Cat No. 70857S; D3F8Q | Cy3 | - | 300 | 5 |
| CD3 | Abcam; Cat No.ab16669; SP7 | Cy5 | - | 189 | 5 |
| CD68 | Abcam; Cat No.ab31630; ED1 | Cy3 | - | 248 | 5 |
| CD45R/B220 | BD Biosciences; Cat No. 553086; RA3-6B2 | Cy5 | - | 200 | 10 |

abscesses and at random in uninfected control groups (see Table 1 group conditions). The size distribution and quantity of *C. auris* abscesses within heart, kidney, and brain tissues was measured using ImageJ. The log base 10 of abscess area ($\mu m^2$ in size) was calculated and graphed using a box and whisker plot to analyze size distribution of fungal abscess for wildtype infected mice (non-neutropenic) and infected neutropenic mice (1A8 and RB6 treated) within each organ tissue. The quantity of abscesses was measured as a count ratio of abscesses within infected tissues to uninfected control groups ("wildtype"). Abscess numbers were pooled for neutropenic mice (both 1A8 and RB6 groups), while also measured separately. Unpaired T-test was used to determine statistically significant differences between groups, with $p<0.05$ marked by and asterisk (*).

QuPath was used to conduct single cell analysis of immune cells within tissues of non-and neutropenic mice. A minimum of 10 ROIs were used to measure T cells (CD3), monocytes (CD68), B cells (B220), and lymphocytes (CD45) and ratio to total immune cell count was calculated. Ratios of immune cells in uninfected non-neutropenic mice (control group) and within *C. auris* abscesses within selected ROIs were calculated. Unpaired T-test was used to determine statistically significant differences between groups, with $p<0.05$ marked by and asterisk (*) and line denoting, which groups were compared. The images used in this analysis are provided in stable repository: https://doi.org/10.5281/zenodo.8144269. Images in the repository are provided as a zipfile, where the OME files can be downloaded and opened with QuPath for further analysis.

## Results

### *C. auris* infection burden in heart, kidney, and brain tissue

Previous work in Torres *et al.* [25] identified heart, kidney, and brain as organs with high *Candida auris* fungal burden in a neutropenic mice model. These tissue samples were thus

prioritized for MxIF analysis (Fig 1), where *C. auris* fungal abscesses were visualized using a polyclonal antibody against *C. albicans* (Thermo Fisher Cat No. PA1-7206), found to cross-react with *C. auris* in culture and in tissues [25]. No commercially available antibody against *C. auris* was identified for use in this study and future immunostaining analysis would benefit from such a resource.

MxIF analysis confirmed these findings, showing, overall, neutropenic mice (both 1A8 and RB6 treated) had a greater number of *C. auris* abscesses within tissue of the heart, kidney, and brain as compared to *C. auris* infected wildtype (non-neutropenic) mice (Figs 2–4). Overall, size of *C. auris* abscess in neutropenic mice was significantly greater (p<0.05).

As expected, the size and number of fungal abscesses were largest in heart and kidney tissues. Area of fungal abscess was significantly larger in the heart (Fig 2) and kidney (Fig 3) of

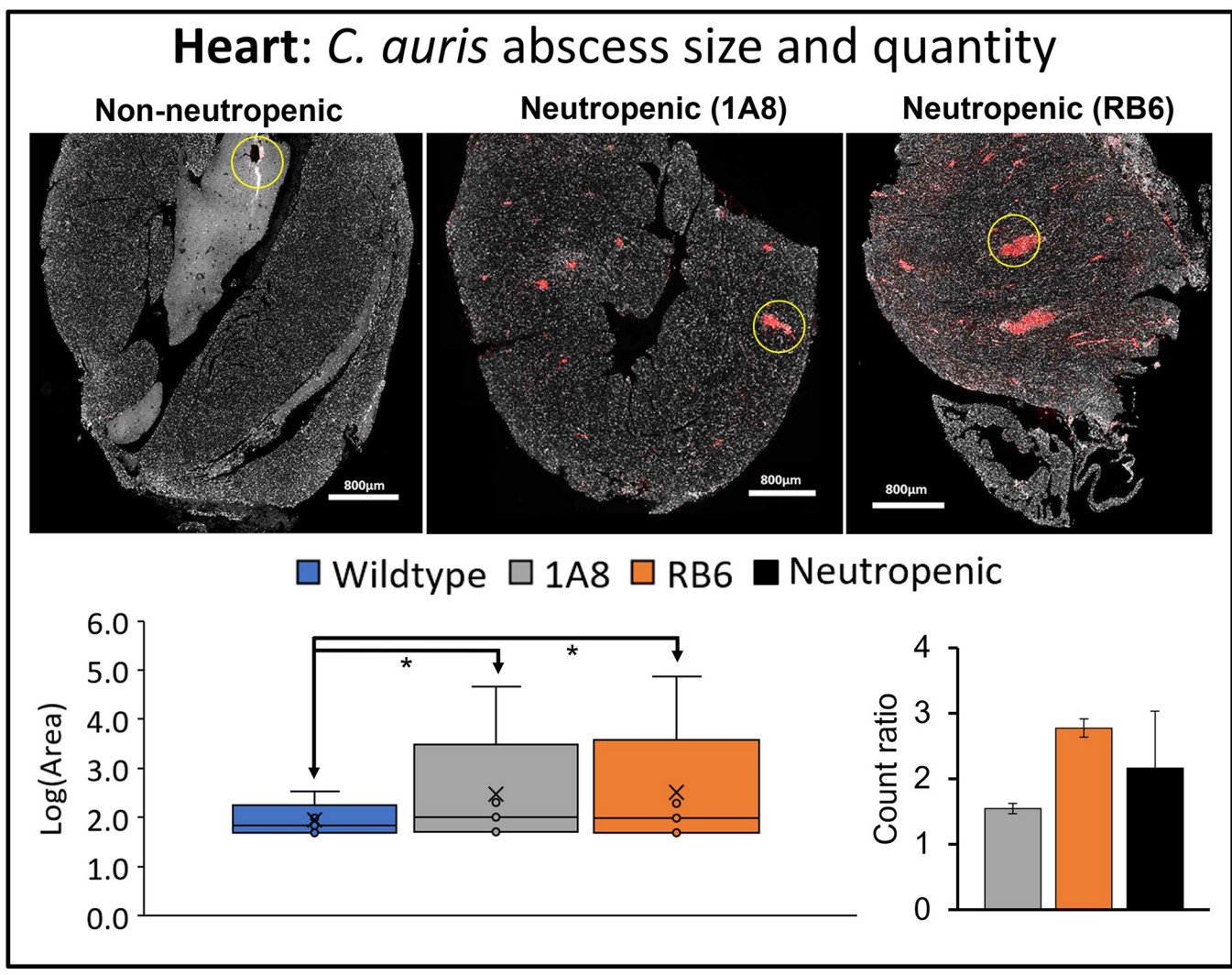

**Fig 2. Infected heart tissue analyzed for size and quantity of *C. auris* abscesses.** Image J analysis of heart tissue from infected wildtype mice ("non-neutropenic") and neutropenic 1A8 and RB6 treated mice. Fungal abscesses are colored in red and an example of *C. auris* fungal abscess is highlighted by a yellow circle. Box and whisker plots show the maximum, minimum, median, and mean log base 10 of *C. auris* fungal abscess area ("Log(Area)"), μm². Quantity of *C. auris* fungal abscesses in each tissue is shown as a "count ratio", where amount of abscess for neutropenic mice is compared to infected non-neutropenic mice. Infected non-neutropenic mice ("Wildtype", blue) and neutropenic mice (1A8, gray; RB6, orange; pooled-1A8 and RB6, black). When found to be significantly different, p<0.05, comparisons between conditions shown by arrowed lines are denoted by an asterisk "*".

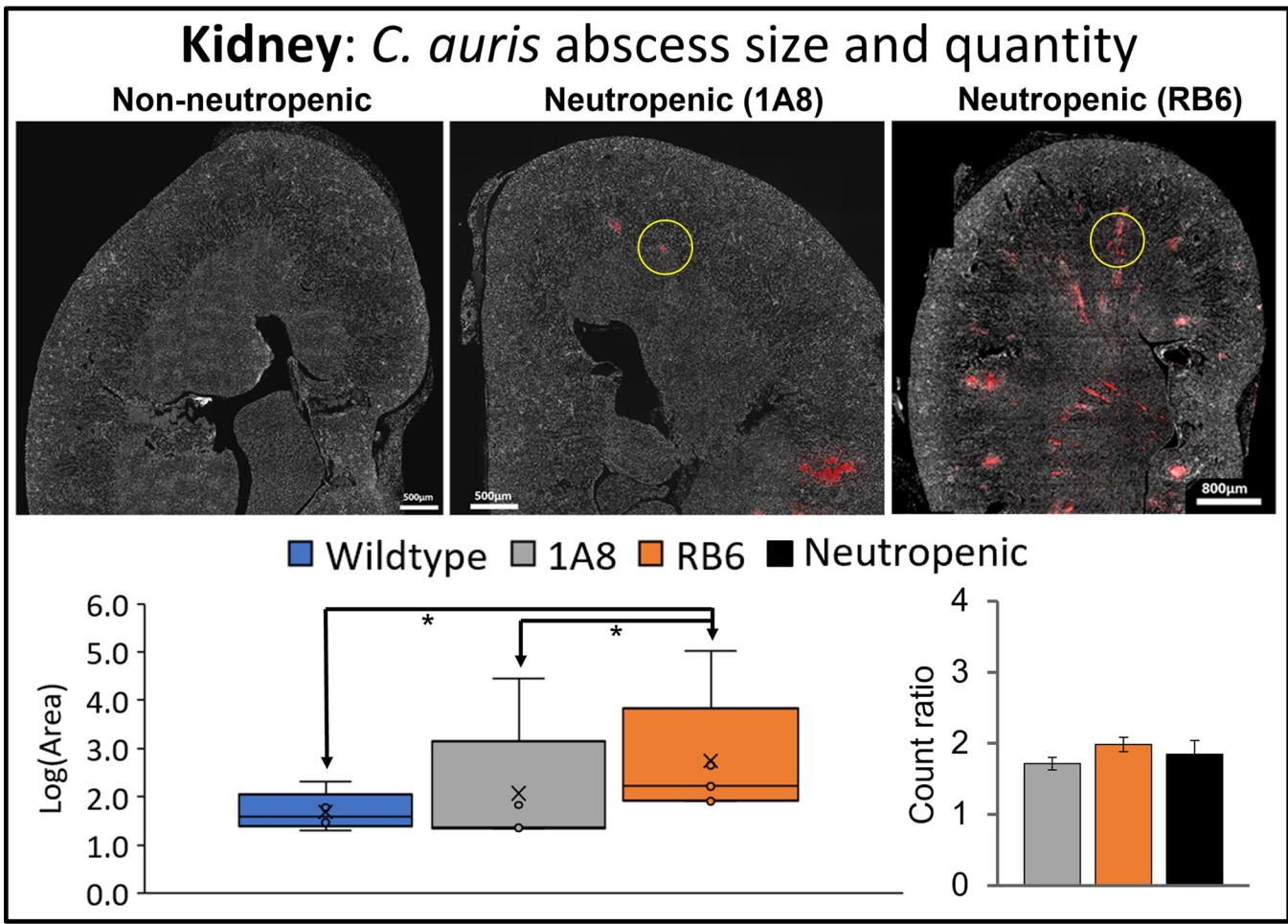

**Fig 3. Infected kidney tissue analyzed for size and quantity of *C. auris* abscesses.** Image J analysis of kidney tissue from infected wildtype mice ("non-neutropenic") and neutropenic 1A8 and RB6 treated mice. Fungal abscesses are colored in red and an example of *C. auris* fungal abscess is highlighted by a yellow circle. Box and whisker plots show the maximum, minimum, median, and mean log base 10 of *C. auris* fungal abscess area ("Log(Area)"), μm². Quantity of *C. auris* fungal abscesses in each tissue is shown as a "count ratio", where amount of abscess for neutropenic mice is compared to infected non-neutropenic mice. Infected non-neutropenic mice ("Wildtype", blue) and neutropenic mice (1A8, gray; RB6, orange; pooled-1A8 and RB6, black). When found to be significantly different, p<0.05, comparisons between conditions shown by arrowed lines are denoted by an asterisk "*".

the neutropenic mice. Size distribution was also larger in all infected organs (including brain) for neutropenic mice (Figs 2–4). Abscesses within heart tissue (Fig 2) were significantly larger (p<0.001) in size in both neutropenic mice models as compared to those in kidney and brain, with the largest quantity of abscesses observed in RB6 mice. In kidney, RB6 treated neutropenic mice models had significantly larger abscesses (nearly 2x greater) compared to infected non-neutropenic (wildtype) controls. In brain tissue, size distribution of *C. auris* abscesses was significant for RB6 neutropenic mice and had a larger number of abscesses as compared to 1A8 mice. Overall, heart and kidney were organs found to be significantly impacted from infection, with RB6 mice being the most impacted.

### Immune response to *C. auris* infection in neutropenic mice

Using MxIF, we analyzed count distribution of monocytes (CD68), lymphocytes (CD45), T cells (CD3), B cells (CD45R/B220), and neutrophils (Ly6G) in heart (Fig 5), kidney (Fig 6) and brain (Fig 7) tissue, see Figs 5–7.

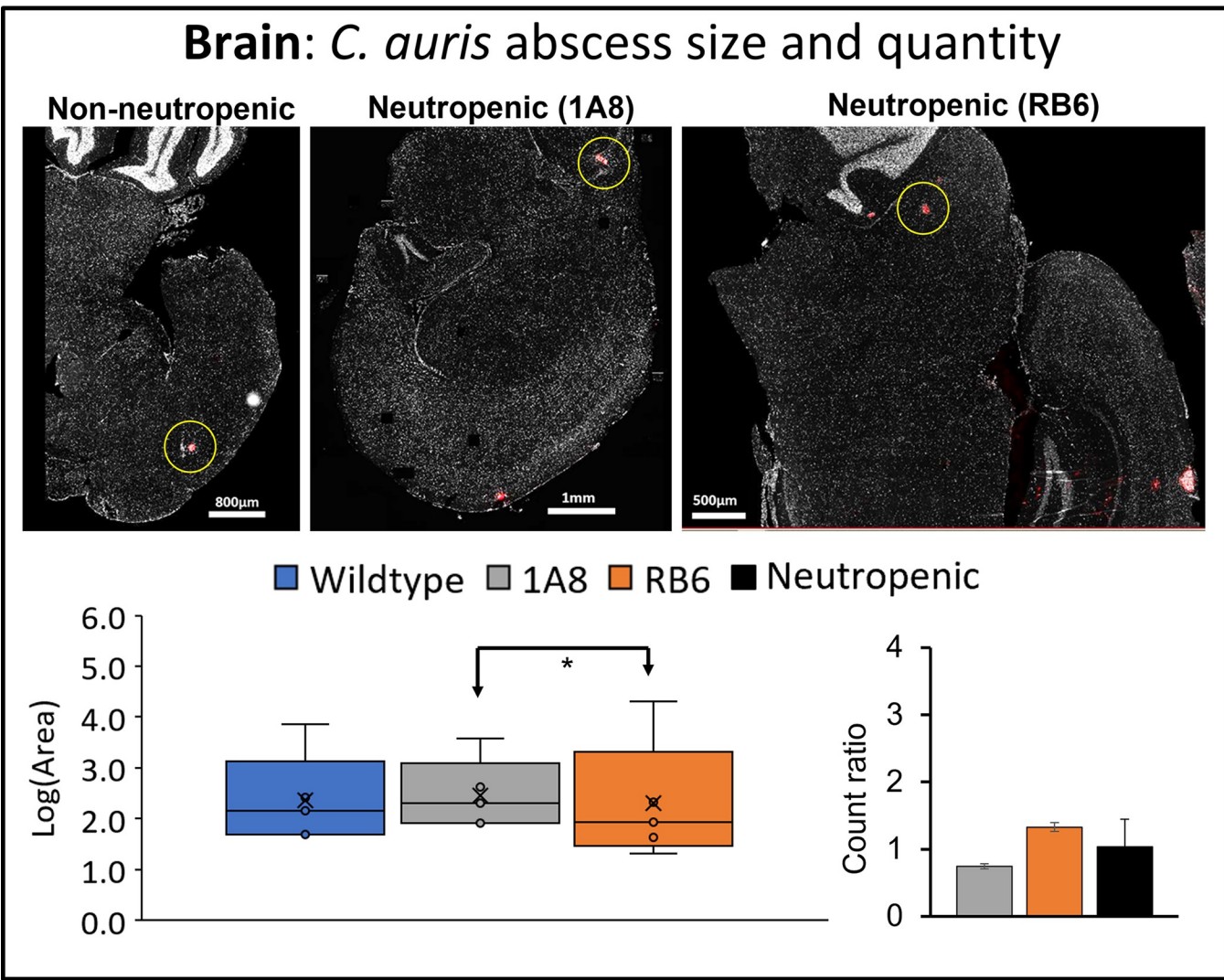

**Fig 4. Infected brain tissue analyzed for size and quantity of *C. auris* abscesses.** Image J analysis of brain tissue from infected wildtype mice ("non-neutropenic") and neutropenic 1A8 and RB6 treated mice. Fungal abscesses are colored in red and an example of *C. auris* fungal abscess is highlighted by a yellow circle. Box and whisker plots show the maximum, minimum, median, and mean log base 10 of *C. auris* fungal abscess area ("Log(Area)"), $\mu m^2$. Quantity of *C. auris* fungal abscesses in each tissue is shown as a "count ratio", where amount of abscess for neutropenic mice is compared to infected non-neutropenic mice. Infected non-neutropenic mice ("Wildtype", blue) and neutropenic mice (1A8, gray; RB6, orange; pooled-1A8 and RB6, black). When found to be significantly different, $p < 0.05$, comparisons between conditions shown by arrowed lines are denoted by an asterisk "*".

MxIF analysis further confirmed previous findings (25), that neutropenic murine models (1A8 or RB6) were depleted of neutrophils and only low background non-specific staining was observed. Ly6G positive cells were detectable in wildtype, non-neutropenic mice ("Non-neutropenic -*C. auris*" and "Non-neutropenic + *C. auris*" in Figs 5–7, bottom MxIF multi-channel and composite image). A significant increase $p < 0.05$) in CD45 cells in heart, kidney, and brain ($p < 0.1$) was also found in infected 1A8 and RB6 neutropenic murine models. In RB6 neutropenic mice, CD45 cells were found to be highest in infected kidney and heart tissue ($p < 0.05$), with highest counts around abscesses (see Fig 6 "Fold Increase" plots). Although CD45 counts were abundant in brain tissue, it was slightly higher than the non-neutropenic *C. auris* infected mice ($p < 0.1$), Fig 7. In addition, many CD45 cells were identified as being T cells (CD3+ CD45+) in heart, kidney, and brain based on co-expression of CD3 and CD45.

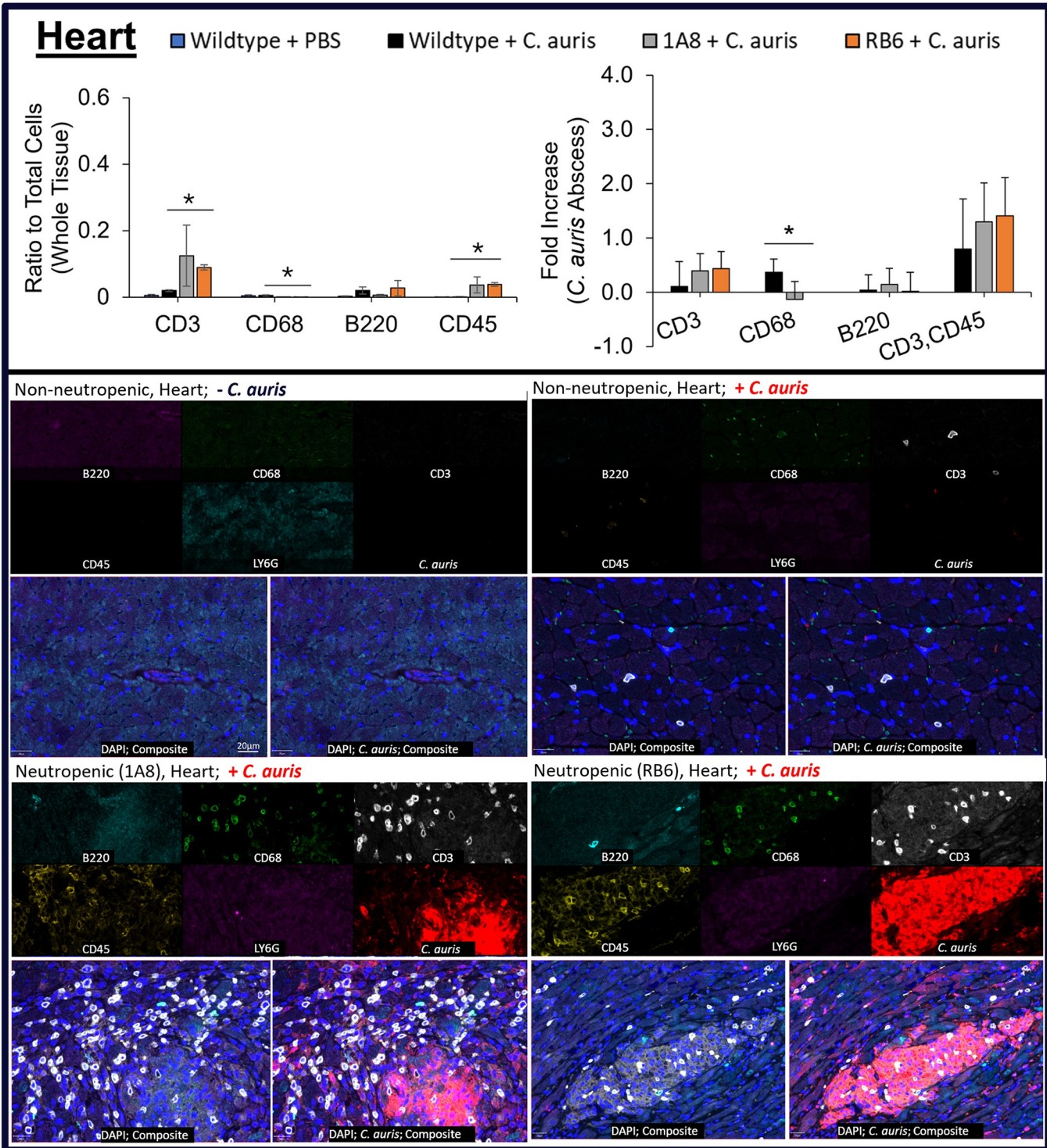

**Fig 5. Immune cell response in heart tissue from *C. auris* infected neutropenic murine models.** Immune cells CD3, CD68, B220, and CD45 (or CD3, CD45) were measured as a ratio to total cells and their fold-increase in heart tissue of uninfected non-neutropenic ("Wildtype + PBS", blue), infected non-neutropenic ("Wildtype + *C. auris*", black), and neutropenic ("1A8 + *C. auris*", gray or "RB6 + *C. auris*", orange) mice. Statistically significant (p<0.05) conditions are denoted by an asterisk (*) with comparisons marked by lines. Below plots, a multi-channel image view of target biomarkers and MxIF composite image (without *C. auris* channel on left and with *C. auris* channel on right) showing B cells (CD45R/B220, turquoise), monocytes (CD68, green), T cells (CD3, white), lymphocytes (CD45, yellow), neutrophils (Ly6G, magenta), *C. auris* (red), and cell nuclei (DAPI, blue).

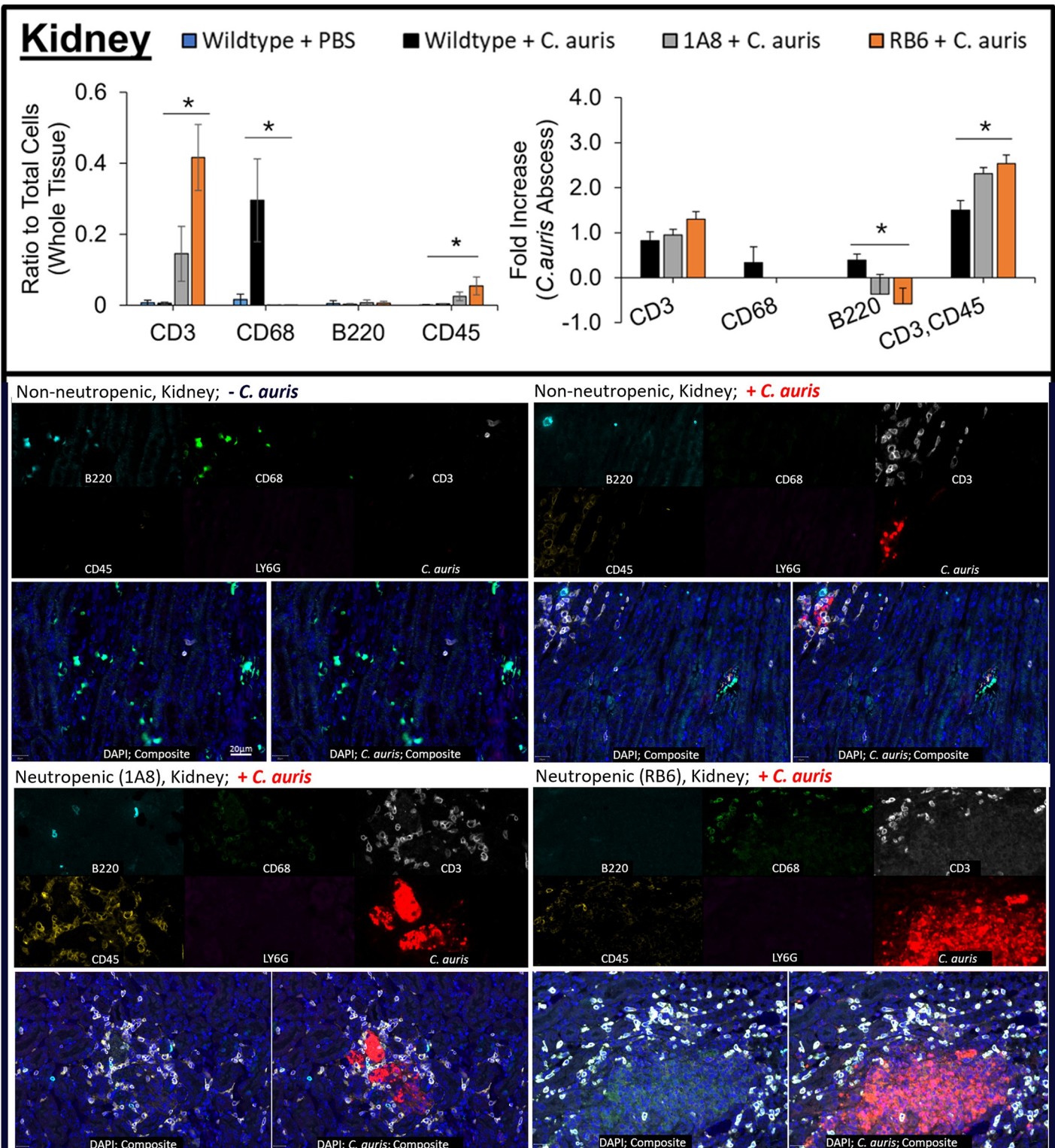

**Fig 6. Immune cell response in kidney tissue from *C. auris* infected neutropenic murine models.** Immune cells CD3, CD68, B220, and CD45 (or CD3, vCD45) were measured as a ratio of total cells and their fold-increase in kidney tissue of uninfected non-neutropenic ("Wildtype + PBS", blue), infected non-neutropenic ("Wildtype + *C. auris*", black), and neutropenic ("1A8 + *C. auris*", gray or "RB6 + *C. auris*", orange) mice. Statistically significant (p<0.05) conditions are denoted by an asterisk (*) with comparisons marked by lines. Below plots, a multi-channel image view of target biomarkers and MxIF composite image (without *C. auris* channel on left and with *C. auris* channel on right) showing B cells (CD45R/B220, turquoise), monocytes (CD68, green), T cells (CD3, white), lymphocytes (CD45, yellow), neutrophils (Ly6G, magenta), *C. auris* (red), and cell nuclei (DAPI, blue).

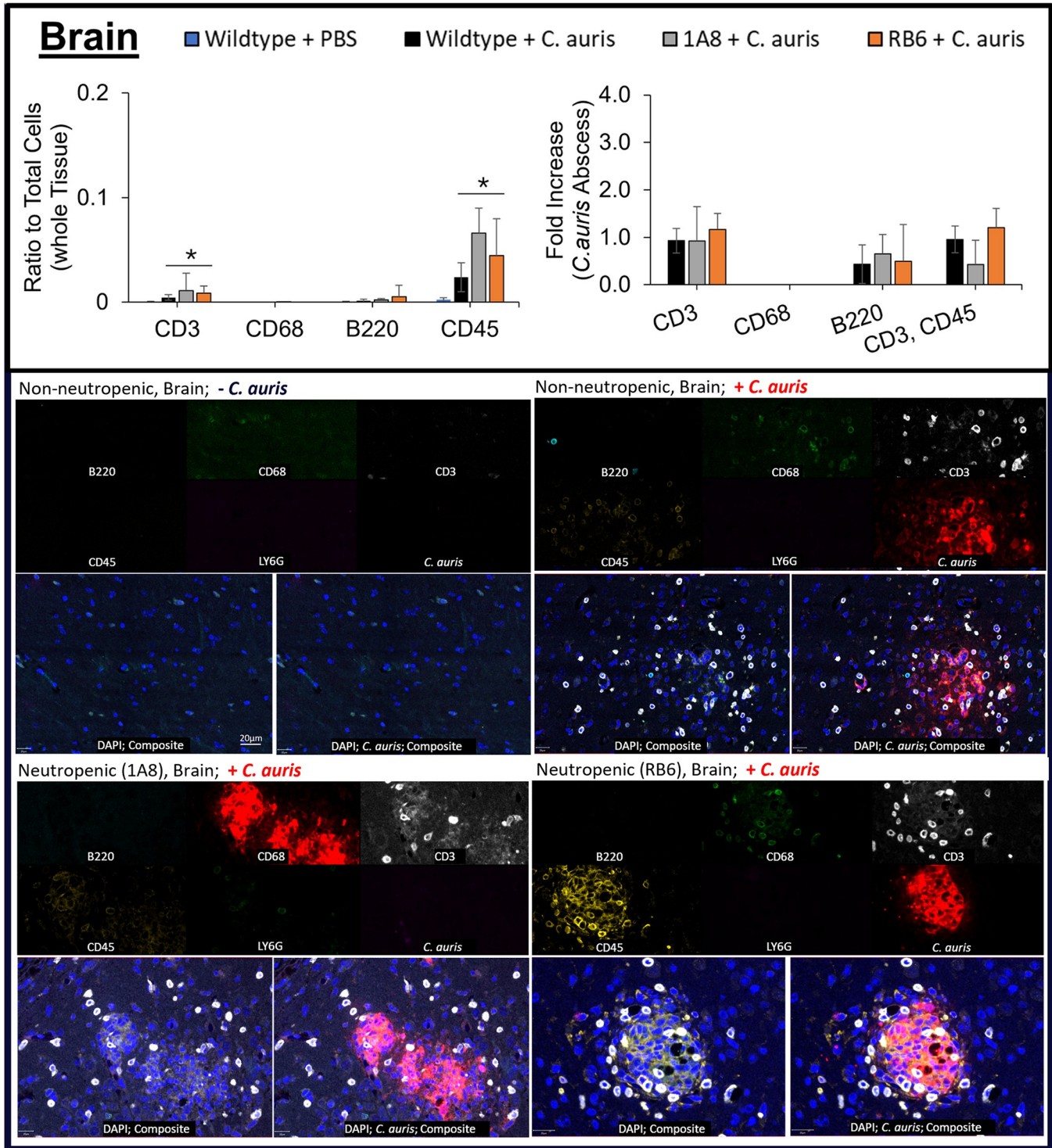

**Fig 7. Immune cell response in brain tissue from *C. auris* infected neutropenic murine models.** Immune cells CD3, CD68, B220, and CD45 (or CD3, CD45) were measured as a ratio to total cells and their fold-increase in brain tissue of uninfected non-neutropenic ("Wildtype + PBS", blue), infected non-neutropenic ("Wildtype + *C. auris*", black), and neutropenic ("1A8 + *C. auris*", gray or "RB6 + *C. auris*", orange) mice. Statistically significant (p<0.1) conditions are denoted by an asterisk (*) with comparisons marked by lines. Below plots, a multi-channel image view of target biomarkers and MxIF composite image (without *C. auris* channel on left and with *C. auris* channel on right) showing B cells (CD45R/B220, turquoise), *C. auris* (red), T cells (CD3, white), lymphocytes (CD45, yellow), monocytes (CD68, green), neutrophils (Ly6G, magenta), and cell nuclei (DAPI, blue).

MxIF analysis showed CD3 T cells to be abundant (p<0.05) in neutropenic mice, especially surrounding the abscess regions. RB6 mice had the highest counts in kidneys (Fig 6). RB6 neutropenic mice had ~49% of their total cells staining for CD3 biomarker in kidney. This was significantly higher (p<0.05) when compared to 1A8 neutropenic mice and even more so when compared to infected, wildtype, non-neutropenic mice. T cell counts for neutropenic mice were 5x higher within heart tissue and >1000x higher in kidney.

Macrophages (CD68 + cells) were most abundant in the kidney of wildtype, non-neutropenic, *C. auris* infected mice and pointedly negligible in neutropenic mice (<0.05% of total cells vs non-neutropenic kidney, 30%, and heart, 0.5%). Similarly, B cells (B220) were most abundant in non-neutropenic mice, but less so in neutropenic mice. The presence of B220 B cells was notable in heart tissue, but only accounted for less than 5% of total cells even within wildtype, non-neutropenic mice, infected with *C. auris*. Neutropenic mice had 100x fewer B cells around abscesses compared to these control mice (p<0.05).

## Discussion and conclusion

MxIF was successfully used to image and analyze *C. auris* infection in neutropenic murine models, showing, at the single cell level, the heterogenous distribution of *C. auris* infection and subsequent immune response by the murine host within heart, kidney, and brain. Our findings support the necessity of the innate immune response, particularly the requirement and activation of neutrophils to be a critical response to fungal infections [30], and is a critical factor in the successful invasiveness *of C. auris* pathology [31, 32]. Thus, we further confirm neutrophils to be critical in the innate immune response against *C. auri*s. MxIF analysis also pointed to the importance of neutrophils in the larger picture of the innate immune response, further highlighting the their potential role in therapeutic and preventative approaches, like antifungal vaccine development dependent on adaptive immune response activation [33]. The role of T cell subtypes such as CD4 and CD8 as well as dendritic cells and deployment of B cells [34] should be further explored to better understand how adaptive immunity can be stimulated to combat *C. auris* infections, especially within immune compromised individuals, such as in neutropenia.

Although, studies point toward the importance of neutrophils in the early stages of an innate immune response, where failure of neutrophils to engage in phagocytose or release NETs contribute to persistence of *C. auris* infections [35], our use of neutropenic murine models show an absence of monocyte (CD68) cells. In contrast, CD68 monocytes, rather than neutrophils (Ly6G), were most abundant (7 days post infection), in the control non-neutropenic mice. This may suggest, long term monocyte mediated defence might play an important role in effective immune responses against fungal infections [36] and may be a critical immunostimulatory component as with other candidiasis [37].

Neutrophil-depleted mice were also observed to have a higher abundance of CD45 lymphocytes, specifically CD3+ T cells, within regions of *C. auris* abscesses. Despite the significant presence of lymphocytes and T cells, neutropenic mice were found to have a larger quantity of notability sized *C. auris* abscesses, suggesting an inability of the adaptive immune response to clear fungal infections in the absence of neutrophils. Our MxIF investigation of *C. auris* infection in neutropenic murine models, further confirmed infection pathology through potential evasion of neutrophils [38] and inactivation of the innate immune response in human hosts. We also found the presence of B cells to be few and independent of neutropenia, which further supports studies that have shown *C. auris* does not elicit an IgG driven B cell response [39]. Since *C. auris* only forms yeast cells, this may suggest B cell response is in fact, more focused toward hyphal forms of invasive yeast [40]. Comparison of B cell response to *C. auris* capable

of filamentation [41] would help broaden our understanding of the yeast's highly adaptive pathophysiology. *C. auris*, unlike other candidiasis, has a morphogenetic plasticity, which would benefit future analytical efforts to investigate how aggregative or non-aggregative phenotypes impact engagement of innate and adaptive immune responses [42–46].

Future analytical efforts would benefit from the use of multiplexed imaging to further characterize immune cell population of the adaptive immunity (CD8+, CD4+, NK Cells) and functional markers of activity and exhaustion. We observed a CD3+ T cell centric immune response in neutropenic mice compared to controls and additional single cell analysis of T cell subtypes across different infected organ tissues can be used to better understand the role of neutrophils in fungal infections. Such analysis maybe able to identify shifts in immune subpopulations throughout infections, helping to develop an immune profile for diagnosis of high-risk patient populations, which may be susceptible to *C. auris* infections. MxIF imaging expands the number of biomarker targets (>60) that more traditional microscopic techniques can analyze in a single tissue specimen [15]. A key element is the use of fluorescent dye chemical inactivation, which allows the reuse of common dyes in subsequent staining and imaging cycles. As a result, a large number of biomarkers can be imaged for analysis using a handful of common fluorescent probes (i.e., FITC, DAPI, CY5, CY3, etc.). Coupled with computational registration of sequential images, a multiplex image with preserved spatial information for use in accurate cellular and subcellular quantification. MxIF imaging provides an advanced method for microscopic imaging, which exponentially increases the number of unique biomarkers that can be used to better characterize disease pathology, such as in fungal infections. In this study, we have shown MxIF imaging capable of analysis the infection pathobiology of *C. auris* at the single cell level in neutropenic mice models.

Overall, we found organ specific immune niches within heart, kidney, and brain tissues infected with *C. auris*, revealing the heterogenous distribution of fungi and a neutrophil driven immune response, which shifts toward a CD3-dominant response in the absence of neutrophils, sustaining abundant and persistent abscesses within infected tissues. This study successfully demonstrated that MxIF may improve our mechanistic understanding of fungal infections, highlighting a possible underlying Th2 driven immune response in neutropenic *C. auris* infection mouse models, which may be leveraged in therapeutic and preventative approaches. We also successfully demonstrated, for the first time, MxIF capability for single cell level analysis of the immune pathology in *C. auris* infection. We showed *C. auris* infected heart, kidney, and brain tissue of neutropenic mice to have fungal abscesses of significant size and quantity as compared to control (infected wildtype, non-neutropenic mice). We also quantified, at the single cell level, the immune profile of different infected organ tissues in comparison to fungal abscesses. Severity of infection was found to be organ specific, with heart and kidney having the greatest fungal burden. Neutropenic mice models treated with RB6, a broader targeting treatment for neutrophil depletion, were also found to have the most significant number and size of *C. auris* abscesses, despite having the highest number of T cells, immune cells with a pivotal role in host's adaptive immunity. Yet, despite the evident shift toward an adaptive immune response, neutropenic mice, are observed to have inadequate clearing of the fungal infection. We find MxIF technology has the potential for documenting the stages of infection at the single cell level, which can be used in early detection diagnostics and for scoring severity of fungal infection. Better understanding of how the immune landscape changes throughout fungal infections can help further development of therapeutics. Increased prevalence of co-infections with of multi-drug resistant fungal organisms (e.g. *C. auris*) with viruses (e.g. COVID-19), and gram-negative bacteria (e.g. *Acinetobacter baumannii*, *Pseudomonas aeruginosa*, and carbapenem resistant enterobacteria, like *Klebsiella pneumoniae*), [8–10] pose a fatal risk to immunocompromised patient populations and warrants a

better of fungal pathobiology. MxIF can give a more in-depth spatial analysis for validation of preventative methods.

## Supporting information

**S1 File. Protocol for antibody characterization.** Protocol document from protocol io (dx.doi.org/10.17504/protocols.io.bpyxmpxn), which outlines antibody characterization for multiplexing procedures.
(PDF)

**S2 File. Protocol for staining and imaging.** Protocol document from protocol io (dx.doi.org/10.17504/protocols.io.bpv6mn9e), which outlines antibody staining and imaging for multiplexing.
(PDF)

**S3 File. Protocol for processing slides for multiplexing.** Protocol document from protocol io (dx.doi.org/10.17504/protocols.io.bpwumpew), which outlines slide clearing and antigen retrieval procedures to prepare slides for multiplex imaging.
(PDF)

## Acknowledgments

We would like to thank the General Electric Company and extend a special thank you to the Wadsworth Center and the University at Albany for their support, through start-up funds, of Dr. Magdia De Jesus and her research team. We also want to acknowledge Dr. Magdia De Jesus research team member, Steven R. Torres, for invaluable technical expertise.

## Author Contributions

**Conceptualization:** Chrystal Chadwick, Magdia De Jesus, Jessica S. Martinez.

**Data curation:** Chrystal Chadwick, Magdia De Jesus, Jessica S. Martinez.

**Formal analysis:** Chrystal Chadwick, Magdia De Jesus, Jessica S. Martinez.

**Funding acquisition:** Magdia De Jesus, Jessica S. Martinez.

**Investigation:** Chrystal Chadwick, Magdia De Jesus, Jessica S. Martinez.

**Methodology:** Chrystal Chadwick, Magdia De Jesus, Jessica S. Martinez.

**Project administration:** Magdia De Jesus, Fiona Ginty, Jessica S. Martinez.

**Resources:** Chrystal Chadwick, Jessica S. Martinez.

**Software:** Chrystal Chadwick, Jessica S. Martinez.

**Supervision:** Magdia De Jesus, Fiona Ginty, Jessica S. Martinez.

**Validation:** Chrystal Chadwick, Magdia De Jesus, Fiona Ginty, Jessica S. Martinez.

**Visualization:** Chrystal Chadwick, Magdia De Jesus, Jessica S. Martinez.

**Writing – original draft:** Chrystal Chadwick, Magdia De Jesus, Fiona Ginty, Jessica S. Martinez.

**Writing – review & editing:** Chrystal Chadwick, Magdia De Jesus, Fiona Ginty, Jessica S. Martinez.

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
