## [Decision Letter · Decision Letter 0]

17 May 2023

PONE-D-23-02779Pathobiology of Candida auris infection analysed by multiplexed imaging and single cell analysisPLOS ONE

Dear Dr. Martinez,

Thank you for submitting your manuscript to PLOS ONE. After careful consideration, we feel that it has merit but does not fully meet PLOS ONE’s publication criteria as it currently stands. Therefore, we invite you to submit a revised version of the manuscript that addresses the points raised during the review process.

This is a valuable contribution to the literature on *Candida auris*. The referees and I have favorably reviewed your manuscript. I will be happy to endorse this work further pending a revision adequately addressing all points raised by both reviewers. Additionally, per PLOS policy please make all raw data available either through a stable repository (recommended) or as a supplement to the manuscript. See applicable policies here: https://journals.plos.org/plosone/s/data-availability. Figures 2-7 present analyzed data for which this applies.

We look forward to receiving your revised manuscript.

Sincerely,

Nicholas A. Pullen, Ph.D.

Academic Editor

PLOS ONE

Journal Requirements:

2. Thank you for stating the following in your Competing Interests section: "No authors have competing interests."

Reviewers' comments:

Reviewer's Responses to Questions

**Comments to the Author**

1. Does the manuscript report a protocol which is of utility to the research community and adds value to the published literature?

Reviewer #1: Yes

Reviewer #2: Yes

2. Has the protocol been described in sufficient detail?

To answer this question, please click the link to protocols.io in the Materials and Methods section of the manuscript (if a link has been provided) or consult the step-by-step protocol in the Supporting Information files.

The step-by-step protocol should contain sufficient detail for another researcher to be able to reproduce all experiments and analyses.

Reviewer #1: Yes

Reviewer #2: Yes

3. Does the protocol describe a validated method?

Reviewer #1: Yes

Reviewer #2: Yes

4. If the manuscript contains new data, have the authors made this data fully available?

Reviewer #1: No

Reviewer #2: Yes

**5. Is the article presented in an intelligible fashion and written in standard English?**

Reviewer #1: Yes

Reviewer #2: Yes

6. Review Comments to the Author

Reviewer #1: The method work "Pathobiology of C. auris infection analysed by multiplexed imaging and single analysis" by Martinez et al. capitalize on previous work by Torres et al., 2020 (PMID: 31818824) in which a neutropenic murine model was developed to study C. auris pathogenesis. Here, the authors tried to characterize C. auris abscesses at cellular level processing tissues collected in the Torres et al, 2020 study and using the MxIF technique. The study provides proof of concept that MxIF can be employed to gain insight into immune responses to C. auris and its pathogenesis. As such it deserves dissemination within the scientific community. However the current submission appears to be hurried up (non-curated bibliography, poor description of statistical methods for quantification, Fig. 5 is the same as Fig. 7 [two duplicated brain figures but no heart one for me to review], poor resolution images) and the work would benefit from a thorough revision.

Major and minor comments:

1) In introduction on line 64 use reference 15 after the term MxIF is introduced for the first time. So the original work of Gerdes et al., 2013 PNAS 110(29):11982-7 is referenced first. This work appears twice in the references list: 15 and 27. Please make sure that all the references are correct and numbered appropriately before resubmission.

2) line 66: is it not "60+" too colloquial?

3) line 103: "scarified sacrificed". Please correct to "sacrificed".

4) lines 124-129: reference supplementary material or add those web links to the references if allowed by the journal format. You did that for Image J and QuPath.

5) line 135: There is a double full stop.

6) line 170: You can just reference (Table 2) and eliminate this "Ly6G and CD45R/B220 at 10 ug/mL, CD45, CD3, and CD68 at 5 ug/mL" The concentrations are listed in Tab.2.

7) Statistical analysis, plots and figures: It is not clear how many abscess were counted and how many serial sections per organ were analyzed. The quantification raw data should be part of supplementary material. Could you please use an overlay of scatter and box plot so that the reader could see the individual data? The area graphs lack a unit of measure which I assume is um^2. If possible you could conceive combining Fig. 2-4 in one picture with letter panels. I could not review Fig. 5 which was supposed to present heart data as Fig. 7 (brain) was accidentally included twice. The resolution of the images is extremely poor, it would be nice for the reviewer to have access to better ones. For Fig 5-7 it would be beneficial to use letter panels.

8) line 375: "andindependent". Include space.

9) line 410: "CD3 dominate". Correct to "CD3-dominant"

10) line 434-436: Remove the abbreviations for the organisms mentioned (e.g. Acinetobacter baumannii (AB)) as they are not used elsewhere in the manuscript. Also the period starting at 432 reads a little odd. Consider rephrasing. I would also mention Carbapenem Resistant Enterobacterales instead of the sole K. pneumoniae.

Reviewer #2: Good to see continued advancement of imaging methods for C. auris. A general comment that the paper provides information both related to the method as well as implications. Given that this paper is classified as a methods paper, suggest providing a bit more context about previous usage of MxIF or similar methods have been used for C. auris or other related yeast, both in mouse models or other. One interesting point is that a C. ablicans antibody was used that is cross-reactive with C. auris. Can the authors comment on any implications or limitations this cross-reactivity and whether a C. auris specific antibiody is available or could be of value? Along similar lines, I think table 2 should clarify that the C. auris target antibody is actually marketted as a C. albicans antibody. Even though this is indicated elsewhere in the text, I think some readers may not catch this so ideal if authors can find a way to incorporate this into the table.

Minor comment

- Line 51 states that CDC has identified 3-5% of all isolates globally are pan-resistant to antifungals, but the associated reference (Ref #3) is not by CDC authors and does not seem relevant. Suggest correcting reference.

7. PLOS authors have the option to publish the peer review history of their article (what does this mean?). If published, this will include your full peer review and any attached files.

Reviewer #1: No

Reviewer #2: No

---

## [Author Response · Author response to Decision Letter 0]

15 Sep 2023

A rebuttal letter has been provided, which addresses all editorial and reviewer questions/feedback.

---

## [Decision Letter · Decision Letter 1]

27 Sep 2023

PONE-D-23-02779R1Pathobiology of Candida auris infection analyzed by multiplexed imaging and single cell analysisPLOS ONE

Dear Dr. Martinez,

Thank you for submitting your manuscript to PLOS ONE. After careful consideration, we feel that it has merit but does not fully meet PLOS ONE’s publication criteria as it currently stands. Therefore, we invite you to submit a revised version of the manuscript that addresses the points raised during the review process.

ACADEMIC EDITOR: Please address new minor concerns. The revised manuscript will be evaluated by editorial team without sending for further review. 

We look forward to receiving your revised manuscript.

Kind regards,

Ashok Kumar, Ph.D.

Academic Editor

PLOS ONE

Journal Requirements:

Additional Editor Comments:

Authors addressed most of the prior concerns. Please address few minor points identified during this round.

Reviewers' comments:

Reviewer's Responses to Questions

**Comments to the Author**

1. Does the manuscript report a protocol which is of utility to the research community and adds value to the published literature?

Reviewer #3: Yes

2. Has the protocol been described in sufficient detail?

To answer this question, please click the link to protocols.io in the Materials and Methods section of the manuscript (if a link has been provided) or consult the step-by-step protocol in the Supporting Information files.

The step-by-step protocol should contain sufficient detail for another researcher to be able to reproduce all experiments and analyses.

Reviewer #3: Yes

3. Does the protocol describe a validated method?

Reviewer #3: Yes

4. If the manuscript contains new data, have the authors made this data fully available?

Reviewer #3: Yes

**5. Is the article presented in an intelligible fashion and written in standard English?**

Reviewer #3: Yes

6. Review Comments to the Author

Reviewer #3: The manuscript has been improved from the original submission. The methods have been explained better. However, there are still typing errors throughout the manuscript and it seems rushed. Also, the fluorescence microscopy images are still not clear. Here are the comments:

1) Line 55: eliminate ‘carries’

2) Line 102: eliminate ‘were’ before sacrificed

3) Lines 115, 272, 285, 297, 309 and any other places: correct to Ly6G and not LyG6

4) Line 149: add a comma between primary secondary.

5) Line 199: replace ‘were’ with ‘where’

6) Line 431: better understanding of..

7) Make sure to write the scientific names in italics throughout the manuscript. One example is in lines 209, 220, 314 etc.

8) The fluorescence image files are not clear for Fig 5, Fig 6 and Fig 7.

7. PLOS authors have the option to publish the peer review history of their article (what does this mean?). If published, this will include your full peer review and any attached files.

Reviewer #3: No

---

## [Author Response · Author response to Decision Letter 1]

29 Sep 2023

Ashok Kumar, Ph.D.

Academic Editor

PLOS ONE

September 29, 2023

Dr. Ashok Kumar and PLOS ONE editorial staff:

In response to academic editor and reviewer responses regarding our revised manuscript (PONE-D-23-02779R1) entitled “Pathobiology of Candida auris infection analyzed by multiplexed imaging and single cell analysis”, by Chrystal Chadwick, Magdia De Jesus, Steven R. Torres, Fiona Ginty, and Jessica S. Martinez, has been further edited. We have outlined each point of concern and annotated how each was addressed in our second revision.

1.) We have made changes to all the minor grammatical/writing corrections outlined by Reviewer 3. 

Line 55: eliminate ‘carries’—was removed.

2) Line 102: eliminate ‘were’ before sacrificed—was removed.

3) Lines 115, 272, 285, 297, 309 and any other places: correct to Ly6G and not LyG6—corrected.

4) Line 149: add a comma between primary secondary. —was added.

5) Line 199: replace ‘were’ with ‘where’—replaced.

6) Line 431: better understanding of.. —line has been corrected.

7) Make sure to write the scientific names in italics throughout the manuscript. One example is in lines 209, 220, 314 etc. —corrected.

2.) Figures 5, 6, and 7 were also revised to provide clear images and we have included larger composite images. Right composite images should all biomarkers, while the left only shows those for immune cells used in the study. Color was also changed for B220 and Ly6G biomarkers and revision was reflected in the figure captions.

We also reviewed references and entire manuscript to ensure it is complete and correct. Our revised figures 5-7 were also reviewed through the Preflight Analysis and Conversion Engine (PACE) digital diagnostic tool to ensure they meet PLOS requirements. We hope you find our second revised manuscript to meet PLOS ONE publication criteria and look forward to your response. 

Sincerely, 

Dr. Jessica Martinez; Lead Scientist

Biosciences, GE Research 

One Research Circle Niskayuna, NY 12309 

Office: 518-387-4506; Mobile: 518-386-9566

E-mail: jessica.s.martinez@ge.com

---

## [Editor Report · Decision Letter 2]

4 Oct 2023

Pathobiology of Candida auris infection analyzed by multiplexed imaging and single cell analysis

PONE-D-23-02779R2

Dear Dr. Martinez,

We’re pleased to inform you that your manuscript has been judged scientifically suitable for publication and will be formally accepted for publication once it meets all outstanding technical requirements.

Kind regards,

Ashok Kumar, Ph.D.

Academic Editor

PLOS ONE
---

## [Editor Report · Acceptance letter]

8 Jan 2024

PONE-D-23-02779R2 

PLOS ONE

Dear Dr. Martinez, 

I'm pleased to inform you that your manuscript has been deemed suitable for publication in PLOS ONE. Congratulations! Your manuscript is now being handed over to our production team.

Kind regards, 

on behalf of

Dr. Ashok Kumar 

Academic Editor

PLOS ONE